# Functional Outcome Analysis of Stereotactic Catheter Aspiration for Spontaneous Intracerebral Hemorrhage: Early or Late Hematoma Evacuation?

**DOI:** 10.3390/jcm12041533

**Published:** 2023-02-15

**Authors:** Yuanjian Fang, Junjie Wang, Luxi Chen, Wei Yan, Shiqi Gao, Yibo Liu, Xiaoyu Wang, Xiao Dong, Jianmin Zhang, Sheng Chen, Fengqiang Liu, Zefeng Wang, Yang Zhang

**Affiliations:** 1Department of Neurosurgery, The Second Affiliated Hospital, Zhejiang University School of Medicine, 88 Jiefang Road, Hangzhou 310009, China; 2Clinical Research Center for Neurological Diseases of Zhejiang Province, Hangzhou 310009, China; 3Department of Neurosurgery, The Fourth Affiliated Hospital, International Institutes of Medicine, Zhejiang University School of Medicine, Yiwu 310030, China; 4Department of Medical Genetics, The Second Affiliated Hospital, Zhejiang University School of Medicine, Hangzhou 310009, China; 5Department of Neurology, Research Center of Neurology, The Second Affiliated Hospital, Zhejiang University School of Medicine, Hangzhou 310009, China

**Keywords:** intracerebral hemorrhage, minimally invasive, stereotactic catheter aspiration, outcome, rebleeding

## Abstract

Background: Minimally invasive stereotactic catheter aspiration becoming a promising surgical alternative for intracerebral hemorrhage (ICH) patients. Our goal is to determine the risk factors that lead to poor functional outcomes in patients undergoing this procedure. Methods: Clinical data of 101 patients with stereotactic catheter ICH aspiration were retrospectively reviewed. Univariate and multiple logistic analyses were used to identify risk factors for poor outcomes 3 months and 1 year after discharge. Univariate analysis was used to compare the functional outcome between early (<48 h after ICH onset) and late hematoma evacuation (≥48 h after ICH onset) groups, as well as for the odd ratios assessment in terms of rebleeding. Results: Independent factors for poor 3-month outcome included lobar ICH, ICH score > 2, rebleeding, and delayed hematoma evacuation. Factors for poor 1-year outcome included age > 60, GCS < 13, lobar ICH, and rebleeding. Early hematoma evacuation was linked to a lower likelihood of poor outcome both 3 months and 1 year post-discharge, but with higher risk of postoperative rebleeding. Conclusions: Lobar ICH and rebleeding independently predicted both poor short- and long-term outcomes in patients with stereotactic catheter ICH evacuation. Early hematoma evacuation with preoperative rebleeding risk evaluation may benefit patients with stereotactic catheter ICH evacuation.

## 1. Introduction

Spontaneous intracerebral hemorrhage (ICH) is a severe type of stroke with high morbidity and mortality. The overall 30-day mortality rate of ICH is reportedly above 30% [1], but only 25% of patients can live independently at 6 months [2]. Intuitively, hematoma is considered one of the primary causes of poor outcomes for patients after ICH. It not only initiates early brain injury via mass effect through direct physical compression but also initiates the second brain injury by blood products and the inflammatory response. Thus, early surgical hematoma evacuation was believed to alleviate the primary and prevent secondary injury after ICH [3]. However, several prospective randomized controlled trials (RCTs) failed to prove the improvement in outcomes in patients treated with conventional surgical evacuation [4,5,6,7,8]. Since most spontaneous ICH is located in deep or lobar brain parenchyma, conventional surgical evacuation may inevitably damage the healthy brain tissue and fiber conduction tracts [9].

Recently, both RCTs [10,11,12] and observational studies from different settings comparing minimally invasive stereotactic catheter aspiration with craniotomy indicated minimally invasive stereotactic catheter aspiration to be a promising surgical alternative for hematoma evacuation in ICH patients, as it could avoid unnecessary surgical brain injury [3,13,14,15]. A meta-analysis based on 12 high-quality RCTs involving 1955 patients revealed that minimally invasive stereotactic catheter aspiration may have increased benefits compared to craniotomy and conservative treatment [16]. Meanwhile, the minimally invasive surgery plus alteplase in the intracerebral hemorrhage evacuation (MISTIE II) trial demonstrated improved efficacy of stereotactic aspiration in patients with ICH [12]. However, it should be mentioned that the appropriate populations for stereotactic catheter aspiration in ICH patients remain controversial. In our study, we aim to identify risk factors for poor functional outcomes in patients undergoing minimally invasive stereotactic catheter ICH aspiration.

## 2. Materials and Methods

### 2.1. Patients Selection

Patients with spontaneous ICH admitted to the Second Affiliated Hospital of Zhejiang University School of Medicine and treated with stereotactic catheter aspiration between 1 January 2021 and 31 July 2021 were reviewed retrospectively. The neurosurgery center of our hospital is the largest center in Zhejiang Province, which performs more than 9000 surgeries every year, with more than 300 surgical ICH evacuations every year.

The inclusion criteria include supratentorial ICH, age > 18 years old, hematoma volume ≥ 20 mL, reduced level of consciousness, muscle force inferior to grade 3, and negative CT angiogram (CTA) findings. Patients were excluded following the criteria of baseline modified Rankin Scale score (mRS) > 1, serious systemic comorbidities prior to ICH onset, or a history of trauma or suspicion for trauma, irreversible coagulopathy, or lost to follow-up.

The Institutional Review Board and Ethics Committee of the Second Affiliated Hospital of Zhejiang University School of Medicine approved all aspects of this study and waived informed consent (No. 2021-0987).

### 2.2. Patient Management

All patients were treated according to the guidelines proposed by the American Heart Association/American Stroke Association Stroke Council [8]. Patients with underlying coagulopathy or a history of anticoagulation and antiplatelet agent usage were treated according to the guidelines proposed by the Neurocritical Care Society and Society of Critical Care Medicine before surgery [17].

The medical history and results of the neurological physical examination were documented at admission. All patients were admitted to the neurosurgical intensive care unit (NICU) during the acute stage, where they received optimal medical treatment and early rehabilitation. CT and CT angiography were performed immediately at admission. Additional CT scans were performed on days 1 and 3. Patients with clinical deterioration also received an immediate CT scan.

### 2.3. Surgical Procedure

Stereotactic catheter aspiration was conducted using either a stereotactic frame (Leksell Vantage, Elekta, Stockholm, Sweden) or frameless stereotactic navigation system (BrainLab AG, Munich, Germany). All surgeries were conducted by a well-trained surgical team (Q.L, Z.W, Y.Z).

Surgical procedures were based on the method described in previous studies [3,18]. Briefly, a thin-section CT scan (1 mm or 1.5 mm) was performed before surgery for the frame parameter or frameless navigation data construction. The puncture point and trajectory were designed along the long axis of the hematoma. The catheter was placed into the hematoma through a sheath into the precalculated depth. The hematoma was gently aspirated using a 10 mL volume syringe at multiple sites along the long axis of the hematoma until resistance was reached. The sheath depth was adjusted to aspirate the remaining hematoma. Saline was used to wash the hematoma cavity until no further blood clots could be aspirated. An additional CT scan was performed after surgery to determine the location of the catheter and the residual hematoma. Next, 50,000 units of urokinase were injected into the hematoma cavity if the residual hematoma was more than 10 mL. The catheter was generally retained for 1–3 days depending on the amount of fluid drainage and the result of repeated CT scans.

### 2.4. Data Collection

The demographic data (age, sex, body mass index (BMI), etc.) and clinical data (Glasgow Coma Scale (GCS), admitting systolic blood pressure, time to evacuation (from ICH onset), and operative time) were reviewed from the inpatient medical record system. Radiological data were assessed using the CT imaging from the radiology image system. Radiological severity of ICH at admission was determined by the ICH score [19], intraventricular hemorrhage (IVH), midline shift, modified Graeb score [20] hematoma location, and the spot sign evaluated by the first CT/CTA scan. Hematoma location was defined as either deep or lobar according to its origination [21].

Preoperative and postoperative hematoma volume was analyzed by the CT scan 3D data set and through the BrainLab or Leksell software as previously described [18]. Evacuation percent was calculated as (postoperative hematoma volume/preoperative hematoma volume) × 100%. Postoperative rebleeding was defined as hematoma volume expansion ≥ 5 mL between any consecutive CT scans.

Data collection was carried out by senior neurosurgeons A.S and S.C. In case of any discrepancy, a third independent neurosurgeon, F.L., conducted a blinded evaluation. The final results were determined through consensus between the two senior neurosurgeons.

### 2.5. Outcome Assessment

Functional outcomes assessed by the modified Rankin Scale (mRS) [22] were followed by telephone, hospital visits, telephone calls, or clinic visits 3 months (short-term) and 1 year (long-term) after discharge. mRS scores ranging from 0 to 3 were considered good outcomes, and those ranging from 4 to 6 were considered poor outcomes [21]. In addition, 30-day mortality, NICU length of stay (LOS), and hospital LOS were also reviewed. Rebleeding was defined as ICH growth > 30% or >6 mL from the baseline ICH volume [3].

## 3. Statistical Analysis

Continuous variables were presented as mean ± standard error (SD) for normal distribution or median with interquartile range for non-Gaussian distribution. Categorical variables were presented as numbers and percentages. Comparisons between groups were performed using the parametric *t*-test for continuous parameters. The Shapiro–Wilk test was used to test the normality. The Kruskal–Wallis test was used for non-Gaussian distribution. The Chi-square test or Fisher’s exact test were used for categorical parameters.

Binary logistic regression analysis with the backward method was performed to evaluate the risk factors for poor outcomes 3 months and 1 year after discharge. Continuous variables, including age, GCS, ICH scores, preoperative volumes, postoperative volumes, and times to evacuation were turned into categorical variables: age > 60, GCS < 13, ICH score > 2, preoperative volume > 50 mL, postoperative volume < 15 mL, and time to evacuation ≥ 48 h for improved interpretation of results. Variables were selected for multivariate logistic regression analysis if a significance level of ≤ 0.20 was observed in the univariate logistic regression analysis. All odds ratio (OR) and 95% confidence interval (CI) in univariate and multivariate logistic regression analyses were recorded.

All *p*-values were two-tailed, and a *p*-value < 0.05 was considered statistically significant. All statistical analyses were performed using SPSS 22.0 (SPSS Institute, Chicago, IL, USA).

## 4. Results

A total of 140 patients with spontaneous ICH treated with stereotactic catheter aspiration were reviewed. Among the 118 supratentorial ICH patients, there were 2 patients with a history suspicious of trauma, 1 patient with serious renal dysfunction before ICH onset, and 14 patients lost to follow-up that were subsequently excluded from the study. Therefore, 101 patients were included for analysis (Figure 1), consisting of 80 (79.2%) males and 31 (20.8%) females. The mean age of the cohort was 58.2 ± 14.0 (range, 25–88 years). Ninety-two (91.1%) patients had a deep hematoma and nine (8.9%) lobar. Sixty-three (62.4%) and thirty-nine (38.6%) patients had an unfavorable outcome 3 months and 1 year after discharge, respectively. Fifty-five (54.5%) patients underwent stereotactic catheter aspiration with the frameless BrainLab navigation system and 46 (45.5%) with the Leksell stereotactic frame. The detailed demographic and clinical data of patients are listed in Table 1.

### 4.1. Risk Factors for Poor Outcome 3 Months after Discharge

Factors associated with poor outcomes 3 months after discharge on univariate analysis are shown in Table 2. Thirty-eight (37.6%) patients had favorable outcomes (mRS 0–3) and sixty-three (62.4%) had poor outcomes (mRS 4–6) 3 months after discharge. The age, GCS score, ICH score, and time to evacuation presented significant statistical differences between the two groups. The mean age of patients in the mRS (4–6) group was significantly higher than the mRS (0–3) group (*p* = 0.008). The patients in the mRS (4–6) group also had a lower GCS score and higher ICH score compared to mRS (0–3) group (*p* = 0.005 and *p* = 0.001, respectively). In addition, the patients in mRS (4–6) group presented a longer time to evacuation compared to the mRS (0–3) group (*p* = 0.044).

In multiple logistic regression analyses (Table 3), factors, including lobar location (OR = 14.061, 95% CI: 1.210–163.413, *p* = 0.035), ICH > 2 (OR = 7.941, 95% CI: 2.074–30.405, *p* = 0.002), time to evacuation ≥ 48 h (OR = 5.661, 95% CI: 2.004–15.955, *p* = 0.001), and rebleeding (OR = 24.776, 95% CI: 2.044–300.292, *p* = 0.012), were shown to be significantly associated with poor outcome 3 months after discharge. To avoid any inaccuracies that may arise secondarily to the inclusion of rebleeding (potential high collinearity with other covariates), another model for adjusting rebleeding was used. In this model, both ICH > 2 (OR = 6.375, 95% CI: 1. 859–21.864, *p* = 0.003) and time to evacuation ≥ 48 h (OR = 3.637, 95% CI: 1.416–9.342, *p* = 0.007) remained statistically significant for predicting poor outcomes 3 months after discharge.

### 4.2. Risk Factors for Poor Outcome 1 Year after Discharge

Factors associated with poor outcomes 1 year after discharge on univariate analysis are depicted in Table 4. Sixty-two (61.4%) patients had favorable outcomes (mRS 0–3) and thirty-nine (38.6%) had poor outcomes (mRS 4–6) 1 year after discharge. The age, anti-coagulation/platelet use, BMI, GCS scores, ICH scores, postoperative volumes, evacuation percentage, and time to evacuation presented significant statistical differences between the two groups. Similarly, the patients in the mRS (4–6) group presented a higher mean age (*p* < 0.001), lower GCS score (*p* < 0.001), higher ICH score (*p* = 0.001), and longer time to evacuation (*p* = 0.031) compared to patients in the mRS (0–3) group. In addition, the patients in mRS (4–6) group also presented a higher incidence of anti-coagulation/platelet use (*p* = 0.020), lower BMI (*p* = 0.009), more postoperative volume (*p* = 0.015), less evacuation percentage (*p* = 0.031), and a higher incidence of rebleeding (*p* = 0.002).

In multiple logistic regression analyses (Table 5), factors including age > 60 (OR = 15.325, 95% CI: 4.369–53.759, *p* = 0.035), GCS < 13 (OR = 27.216, 95% CI: 4.181–177.154, *p* = 0.001), lobar location (OR = 24.974, 95% CI: 1.400–445.350, *p* = 0.029), and rebleeding (OR = 81.357, 95% CI: 5.453–1213.771, *p* = 0.012) were shown to be significantly associated with poor outcome 1 year after discharge. Meanwhile, the age > 60 (OR = 14.758, 95% CI: 4. 423–49.247, *p* < 0.001) and GCS < 13 (OR = 6.042, 95% CI: 1.466–24.896, *p* = 0.013) remained statistically significant for predicting poor outcome 1 year after discharge in the adjusted model.

### 4.3. Comparison of Patients with Early Hematoma Evacuation and Late Hematoma Evacuation

Univariate analysis between patients who underwent hematoma evacuation < 48 h (early hematoma evacuation) and ≥ 48 h are shown in Table 6. Forty-four (43.6%) patients underwent hematoma evacuation < 48 h, and fifty-seven (56.4%) patients underwent hematoma evacuation ≥ 48 h. Despite a lack of significant difference found in demographic data, clinical data, and device used between the two groups (both *p* > 0.05), the patients with early hematoma evacuation were less likely to have poor outcomes (mRS 4–6) 3 months and 1 year after discharge (*p* = 0.008 and *p* = 0.040, respectively). However, it should be mentioned that the postoperative rebleeding rate was significantly higher in patients with early hematoma evacuation when compared to the late hematoma evacuation group (20.5% versus 7.0%, *p* = 0.046).

### 4.4. Factors for Predicting Postoperative Rebleeding

The ORs assessed by the univariate analysis between patients with and without postoperative rebleeding are shown in Figure 2. Factors including age > 60, female, BMI < 24, anticoagulation or antiplatelet use, spot sign, lobar location, postoperative volume, evacuation percentage, and ≤15 mL residual were significantly associated with postoperative rebleeding (both *p* < 0.05). Notably, the anticoagulation or antiplatelet use (OR = 19.111, 95% CI: 3.063–119.257, *p* = 0.002), lobar location (OR = 7.378, 95% CI: 1.673–32.535, *p* = 0.015), >15 mL residual (OR = 8.049, 95% CI: 1.882–34.428, *p* = 0.001), and spot sign (OR = 5.495, 95% CI: 0.917–32.918, *p* < 0.001) presented as the top four ORs for predicting postoperative rebleeding.

## 5. Discussion

Recently, minimally invasive stereotactic catheter aspiration was recognized as a promising surgical choice for ICH patients. Several RCTs have proven its benefits in both short- [21] and long-term outcomes [23] after ICH. However, the results of MISTIE III showed that the minimally invasive stereotactic catheter aspiration did not improve long-term outcomes 1 year after discharge [24]. The variation in outcomes for ICH patients undergoing minimally invasive stereotactic catheter aspiration may be due to differences in patient characteristics and the choice of surgical navigation system, as well as the timing of hematoma evacuation. Thus, it is important to determine the factors that affect outcomes in these patients.

Our study found that several factors were associated with poor outcomes in ICH patients undergoing minimally invasive stereotactic catheter aspiration. These factors included lobar ICH location, high ICH score, postoperative rebleeding, and late hematoma evacuation (≥48 h after ICH onset) short-term (3 months) and long-term (1 year) after discharge. On the other hand, early hematoma evacuation (<48 h after ICH onset) was linked to lower odds of poor outcomes, but with a higher risk of postoperative rebleeding. Additionally, other risk factors for postoperative rebleeding were identified, such as a history of anticoagulation/antiplatelet use, a spot sign on imaging, a residual hematoma volume of over 15 mL, lobar ICH location, female sex, and a BMI below 24.

The MISTIE III trial cohort analysis showed that factors such as lobar or deep ICH location, age, and presentation GCS score have a significant independent impact on patient outcomes 1 year after discharge [25], aligning with our findings. Additionally, we were the first to demonstrate that factors such as lobar or deep ICH location, ICH score, and time to evacuation play a crucial role in predicting short-term outcomes 3 months after discharge, surpassing the significance of age and GCS score.

The lobar ICH independently predicted both short- and long-term outcomes of patients with minimally invasive stereotactic catheter aspiration in our study. Furthermore, patients with lobar ICH also have a higher incidence of postoperative rebleeding when compared to deep ICH. This result was consistent with the meta-analysis based on 122 studies [26]. They also revealed that the ICH recurrence was higher after lobar ICH than after non-lobar ICH, which was possibly caused by underlying cerebral amyloid angiopathy, microaneurysm rupture, or variation in the use of antihypertensive therapy [26].

The ICH score is a simple clinical grading scale that is calculated using GCS score, age, infratentorial ICH, ICH volume, and IVH [19]. While the ICH score was first created for predicting short-term mortality and outcome, the efficacy of the ICH score may be reduced when predicting the prognosis of functional long-term outcomes [27]. In contrast, the GCS score was designed to reflect the level of consciousness on hospital admission and the initial brain damage. Numerous risk models have demonstrated high accuracy in predicting long-term outcomes [5,28,29], which explained that the ICH score showed a significant association with short-term outcomes and the GCS score showed a significant association with long-term outcomes in our study.

Age was a controversial factor in predicting outcomes as per several studies. While age has been proven as an independent predictor of ICH outcome in some prediction models [19,21,25], it has also been shown to have no significant independent association with outcomes in other studies [30,31]. This discrepancy may result from the different general physical conditions of the elderly and the overall medical care decisions for the elderly in various centers (aggressive or conservative) [19]. In our study, age demonstrated independent significance for predicting long-term outcomes rather than short-term outcomes. We believe the vulnerability of elders in long-term recovery to disability-related complications and loss of rehabilitation training may be due to financial burden on the family.

Of note, the time to evacuation remains a topic of discussion in the surgical management of spontaneous ICH. Early evacuation may reduce not only hematoma volume but also secondary brain injury. Several studies have demonstrated that early evacuation via minimally invasive surgery improves outcomes [21,32]. A meta-analysis analyzing 12 surgical ICH RCTs also found a significant relationship between early time to surgical evacuation and improved outcomes [33]. An observation study based on 90 spontaneous ICH patients with minimally invasive endoscopic ICH evacuation < 24 h showed that time to evacuation (per hour; OR, 0.95 [95% CI, 0.92–0.98]) was significantly related to the 1-year long-term outcome [21]. In our study, early evacuation (within 48 h) was not found to be associated with an increased risk of rebleeding. However, the incidence of rebleeding was higher in patients who underwent early evacuation compared to those who underwent late evacuation. The study showed that this higher incidence was due to several risk factors present in early evacuation cases. Two patients in the early evacuation group had a history of aspirin use, which had been suspended less than 2 days prior. This increased the risk of rebleeding. Additionally, two patients in the early evacuation group had a “spot sign” on their initial CT scan, which is another risk factor for rebleeding. Furthermore, two patients in the early evacuation group had lobar ICH, which is also a risk factor for rebleeding. In contrast, the late evacuation group had one patient who was taking warfarin (with an INR of 1.4 after medical adjustment), one patient with a spot sign on their initial CT scan, one patient with lobar ICH, and one patient with a spot sign on their initial CT scan. While these risk factors were present in the late evacuation group, they still had a lower incidence of rebleeding compared to the early evacuation group.

Postoperative rebleeding is a wild-recognized poor outcome indicator [1]. There were 13 (12.9%) patients that had suffered postoperative rebleeding, which is lower than the MISIE trial (22.2%) [12]. However, another retrospective cohort study based on 142 patients performed with minimally invasive stereotactic catheter aspiration showed only a 4.2% rebleeding rate. All patients with rebleeding in their study were accompanied by basal ganglia ICH, and three had spot signs on the CT scan [3]. In our study, only nine patients with deep ICH had postoperative rebleeding (9/92, 9.8%), with three having spot signs on CT scan and three having anticoagulation/antiplatelet use.

With the advent of augmented reality/mixed reality techniques, new frameless navigation products, such as the BrainLab system, have been recently used as a faster and more feasible technique for minimally invasive hematoma evacuation in ICH patients. These were proven to have a close accuracy for intracranial puncture when compared with the current conventional frame-based stereotactic puncture in an intracerebral hemorrhage phantom model [34]. In our study, we used both Leksell Vantage stereotactic frame system and BrainLab frameless stereotactic system for hematoma drainage. No statistical difference was found in short- and long- term outcomes, as well as rebleeding, which also indicated the feasibility of the BrainLab system in the minimally invasive stereotactic catheter aspiration. However, more studies with large patient cohorts are encouraged to validate the accuracy and efficiency of the BrainLab system in the use of minimally invasive stereotactic catheter aspiration after ICH.

Several limitations are present in this study. Firstly, it is a retrospective, single-center analysis relying on medical records, and those patients who were lost to follow-up were excluded, which may not accurately reflect the patient conditions and bring some potential bias to the results. Additionally, all surgeries were performed by one surgical team, limiting the generalizability of the results. Secondly, there was no medical control group to compare the benefits of surgical intervention and potential confounding factors present. Lastly, the definition of “early evacuation” in the literature varies widely, ranging from 4 to 24 h [21,35]. Our study defined evacuation within 48 h as early evacuation according to our patient cohort, which presented no statistical difference in baseline characteristics between the two groups. However, prospective studies evaluating minimally invasive evacuation in this early time window are needed.

## 6. Conclusions

Lobar ICH and postoperative rebleeding are able to independently predict poor short- and long-term outcomes in patients with minimally invasive stereotactic catheter ICH evacuation. In addition, ICH score > 2 and late hematoma evacuation independently predicted poor short-term outcomes, with age > 60 years-old and GCS < 13 predicting poor long-term outcomes. Early hematoma evacuation presented a lower likelihood of poor outcome both 3 months and 1 year after discharge, although with a higher incidence of postoperative rebleeding, which warrants further study of large, prospective, randomized trials for evaluating the best time window to perform the minimally invasive stereotactic catheter ICH evacuation. A better understanding of the surgical time window for stereotactic catheter ICH evacuation and evaluation of preoperative rebleeding risk may benefit patients with ICH.

## Figures and Tables

**Figure 1 jcm-12-01533-f001:**
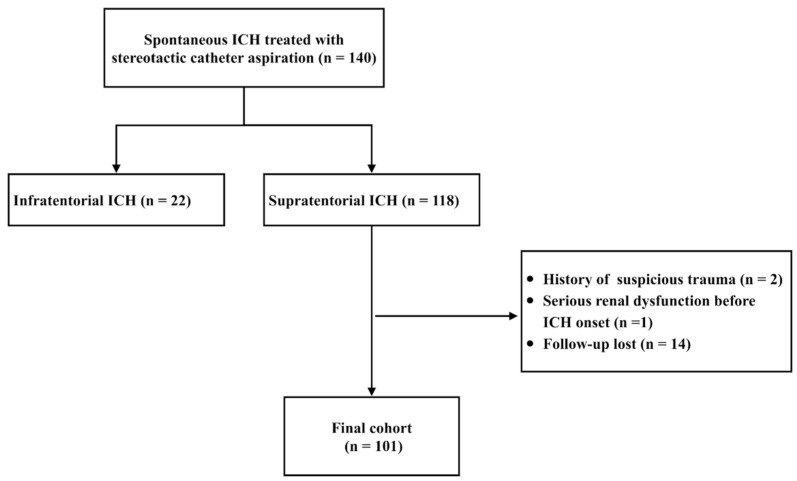
Flow chart of patients’ selection.

**Figure 2 jcm-12-01533-f002:**
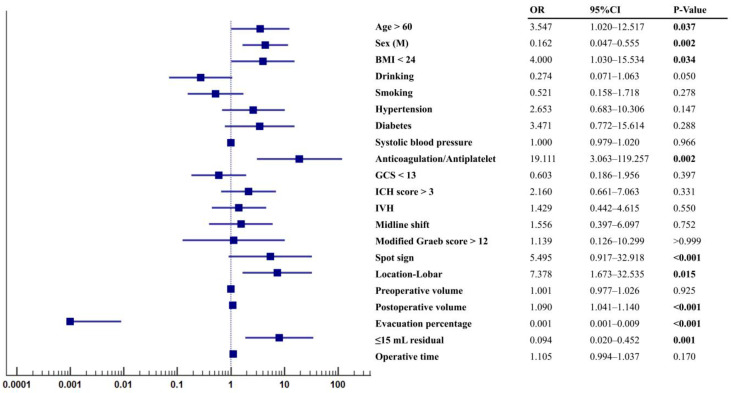
Odd ratios for patients with rebleeding. Statistic differences were analyzed by univariate regression analysis.

**Table 1 jcm-12-01533-t001:** Demographic and Clinical Data.

	Average/N	SD/%
Age, y	58.2	14.0
Sex (M)	80	79.2
BMI	24.5	4.3
Drinking	49	48.5
Smoking	53	52.5
Hypertension	59	58.4
Diabetes	10	9.9
Systolic blood pressure, mmHg	158.1	29.0
Anticoagulation/Antiplatelet	6	6.0
GCS (3–15)		
13–15	36	35.6
7–12	54	53.5
3–7	11	10.9
ICH score		
0–2	31	30.7
3–6	80	69.3
IVH	39	38.6
Midline shift	70	69.3
Modified Graeb score > 12	7	6.9
Spot sign	6	5.9
Location		
Deep	92	91.1
Lobar	9	8.9
Device		
BrainLab	55	54.5
Leksell	46	45.5
Preoperative volume, mL	54.1	23.7
Postoperative volume, mL	16.9	15.2
Evacuation percentage %	70	21.7
≤15 mL residual	60	59.4
Time to evacuation, h	64.1	59.7
Operative time, h	60.6	22.6
Postoperative rebleeding	13	12.9
NSICU LOS, d	5.6	5.3
LOS, d	12.5	6.0
30-day mortality	7	6.9
3-month mRS (4–6)	63	62.4
1-year mRS (4–6)	39	38.6

**Table 2 jcm-12-01533-t002:** Univariate analysis of poor outcome 3 months after discharge.

	mRS 0–3 (*n* = 38) *	mRS 4–6 (*n* = 63) *	*p*-Value ^#^
Demographic data			
Male	31 (81.6)	49 (77.8)	0.648
Age	53.4 ± 11.9	61.0 ± 11.4	**0.008**
Drinking	19 (50.0)	30 (47.6)	0.817
Smoking	22 (57.9)	31 (49.2)	0.397
Hypertension	18 (47.4)	41 (65.1)	0.080
Diabetes	3 (7.9)	7 (11.1)	0.856
Anti-coagulation/platelet	0 (0)	6 (9.5)	0.127
BMI	24.9 ± 4.8	24.2 ± 3.9	0.385
Clinical data			
Systolic blood pressure	153.5 ± 25.6	160.6 ± 30.6	0.228
GCS	12.0 ± 2.8	10.2 ± 3.1	**0.005**
ICH score	1.6 ± 0.8	2.3 ± 1.0	**0.001**
IVH	11 (28.9)	28 (44.4)	0.121
Midline shift	24 (63.2)	46 (73.0)	0.298
Modified Graeb score > 12	3 (7.9)	4 (6.3)	0.767
Spot sign	2 (5.3)	4 (6.3)	>0.999
Location: Deep	33 (86.8)	59 (93.7)	0.422
Preoperative volume, mL	50.2 ± 22.7	56.6 ± 24.1	0.197
Postoperative volume, mL	14.1 ± 11.6	18.4 ± 16.8	0.170
Evacuation percentage			
≤15 mL residual	23 (60.5)	37 (58.7)	0.859
Time to evacuation, h	48.8 ± 38.5	73.4 ± 67.7	**0.044**
Operative time, h	61.0 ± 21.8	60.4 ± 23.1	0.902
Postoperative rebleeding	2 (5.3)	11 (17.5)	0.142
Device			
BrainLab	19 (50.0)	36 (57.1)	0.485
Leksell	19 (50.0)	27 (42.9)	

* Data presented by number (%) or average ± SD. ^#^ Statistic differences were analyzed by Chi-square test, Fisher’s test, and Student’s *t* test.

**Table 3 jcm-12-01533-t003:** Multivariate analysis of poor outcome 3 months after discharge.

				Adjusted for Rebleeding		
Independent Variables	OR	95% CI	*p*-Value	OR	95% CI	*p*-Value
Location: Lobar	14.061	1.210–163.413	0.035			
ICH score > 2	7.941	2.074–30.405	0.002	6.375	1.859–21.864	0.003
Time to evacuation ≥ 48 h	5.661	2.004–15.955	0.001	3.637	1.416–9.342	0.007
Rebleeding	24.776	2.044–300.292	0.012			

**Table 4 jcm-12-01533-t004:** Univariate analysis of poor outcome 1 year after discharge.

	mRS 0–3 (*n* = 62) *	mRS 4–6 (*n* = 39) *	*p*-Value ^#^
Demographic data			
Male	52 (83.9)	28 (71.8)	0.145
Age	52.4 ± 10.7	67.3 ± 13.9	**<0.001**
Drinking	31 (50.0)	18 (46.2)	0.707
Smoking	38 (61.3)	15 (38.5)	0.025
Hypertension	35 (56.5)	24 (61.5)	0.614
Diabetes	5 (8.1)	5 (12.8)	0.436
Anti-coagulation/platelet	1 (1.6)	5 (12.8)	**0.020**
BMI	25.3 ± 4.4	22.1 ± 3.7	**0.009**
Clinical data			
Systolic blood pressure	156.8 ± 29.2	159.7 ± 28.7	0.635
GCS	11.7 ± 2.9	9.5 ± 3.0	**<0.001**
ICH score	1.8 ± 0.8	2.4 ± 1.1	**0.001**
IVH	23 (37.1)	16 (41.0)	0.693
Midline shift	43 (69.4)	27 (69.2)	0.989
Modified Graeb score > 12	6 (9.7)	1 (2.6)	0.171
Spot sign	4 (6.5)	2 (5.2)	>0.999
Location: Deep	57 (91.9)	35 (89.7)	0.707
Preoperative volume, mL	51.8 ± 22.3	57.8 ± 25.5	0.225
Postoperative volume, mL	11.4 (5.6–25.0)	12.4 (6.0–25.9)	**0.015**
Evacuation percentage	74.7 (58.1–85.4)	73.7 (57.0–85.0)	**0.031**
≤15 mL residual	40 (64.5)	20 (51.3)	0.187
Time to evacuation	54.8 ± 39.4	80 ± 80.9	**0.039**
Operative time	60.0 ± 19.7	61.6 ± 26.6	0.726
Postoperative rebleeding	3 (4.8)	10 (25.6)	**0.002**
Device			
BrainLab	29 (46.8)	26 (66.7)	0.052
Leksell	33 (53.2)	13 (33.3)	

* Data presented by number (%) or average ± SD or median (interquartile range). ^#^ Statistic differences were analyzed by Chi-square test, Fisher’s test, Student’s *t* test, and Kruskal–Wallis test.

**Table 5 jcm-12-01533-t005:** Multivariate analysis of poor outcome 1 year after discharge.

				Adjust for Rebleeding		
Independent Variable	OR	95% CI	*p*-Value	OR	95% CI	*p*-Value
Age > 60	15.325	4.369–53.759	<0.001	14.758	4.423–49.247	<0.001
GCS < 13	27.216	4.181–177.154	0.001	6.042	1.466–24.896	0.013
Location-Lobar	24.974	1.400–445.350	0.029			
Rebleeding	81.357	5.453–1213.771	0.001			

**Table 6 jcm-12-01533-t006:** Univariate analysis for patients performed surgery within 48 h and beyond 48 h.

	<48 h (*n* = 44) *	≥48 h (*n* = 57) *	*p*-Value ^#^
Demographic data			
Male	35 (79.5)	45 (78.9)	0.941
Age, y	55.1 ± 13.7	60.5 ± 13.9	0.054
Drinking	22 (50.0)	27 (47.4)	0.793
Smoking	25 (56.8)	28 (49.1)	0.443
Hypertension	28 (63.6)	31 (54.4)	0.350
Diabetes	3 (6.8)	7 (12.3)	0.362
Anticoagulation/Antiplatelet	3 (6.8)	3 (5.3)	0.743
BMI	25.2 ± 4.4	23.9 ± 4.1	0.143
Clinical data			
Systolic blood pressure	161.5 ± 30.8	155.2 ± 27.3	0.284
GCS (3–15)	11.6 ± 3.3	10.4 ± 2.9	0.054
ICH score (0–6)	1.9 ± 0.9	2.2 ± 1.0	0.133
IVH	20 (45.5)	19 (33.3)	0.165
Midline shift	27 (61.4)	43 (75.4)	0.128
Modified Graeb score > 12	6 (13.6)	1 (1.8)	0.053
Spot sign	4 (9.1)	2 (3.5)	0.452
Location: Deep	40 (90.9)	52 (91.2)	0.956
Preoperative volume, mL	51.9 ± 23.6	55.5 ± 23.8	0.407
Postoperative volume, mL	17.6 ± 13.7	16.1 ± 16.2	0.639
Evacuation percentage	66.3 ± 24.4	73.0 ± 19.0	0.120
≤15 mL residual	23 (52.3)	37 (64.9)	0.200
Operative time, h	63.3 ± 22.9	58.5 ± 2.2	0.292
Postoperative rebleeding	9 (20.5)	4 (7.0)	**0.046**
Outcome			
NSICU LOS, d	5.2 ± 4.3	6.0 ± 5.9	0.474
LOS, d	11.7 ± 5.5	13.2 ± 5.3	0.214
30-day mortality	3 (6.8)	4 (7.0)	0.975
3-months mRS (4–6)	21 (47.7)	42 (73.7)	**0.008**
1-year mRS (4–6)	12 (27.3)	27 (47.4)	**0.040**
Device			
BrainLab	26 (59.1)	29 (50.9)	0.411
Leksell	18 (40.9)	28 (49.1)	

* Data presented by number (%) or average ± SD. ^#^ Statistic differences were analyzed by Chi-square test, Fisher’s test, and Student’s *t* test.

## Data Availability

The data presented in this study are available on request from the corresponding author.

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
