# Peer review of "Functional Outcome Analysis of Stereotactic Catheter Aspiration for Spontaneous Intracerebral Hemorrhage: Early or Late Hematoma Evacuation?"

_jcm, 2023, doi:10.3390/jcm12041533_

Round 1
Reviewer 1 Report
Comment-1
I would like to suggest that the author should represent the results in graphical form rather than the tabular form. The missing part is that it is difficult to understand the results without seeing them in graphs. The author can represent the results in graphs.
The study needs more data to prove that stereotactic catheter aspiration becoming a promising 25 surgical alternative for intracerebral hemorrhage (ICH) patients. Also the new techniques developed in ICH were not discussed. Future directions of ICH surgery should be included in the manuscript.
The manuscript is weak in terms of data collection and representation.
Author Response
Reviewer 1
- I would like to suggest that the author should represent the results in graphical form rather than the tabular form. The missing part is that it is difficult to understand the results without seeing them in graphs. The author can represent the results in graphs.
A: Thank your suggestion, we have changed Table 7 into a graphical form for a better understanding of the results, please see Fig 2.
- The study needs more data to prove that stereotactic catheter aspiration becoming a promising surgical alternative for intracerebral hemorrhage (ICH) patients. Also, the new techniques developed in ICH were not discussed. Future directions of ICH surgery should be included in the manuscript.
A: Thank you for your advice.
- We have introduced the advantage of stereotactic catheter aspiration for intracerebral hemorrhage (ICH) patients in the second paragraph of the Introduction. We also added several RCTs (ref. 10-12) and retrospective studies (ref. 14 and 15) to support the use of minimally invasive stereotactic catheter aspiration in the surgical treatment of intracerebral hemorrhage (ICH) patients.
Please see Page 2, Line 78-82
“Recently, both RCTs[10-12] and observational studies from different settings com-paring minimally invasive stereotactic catheter aspiration with craniotomy indicated minimally invasive stereotactic catheter aspiration to be a promising surgical alternative for hematoma evacuation in ICH patients, as it could avoid unnecessary surgical brain injury[3,13-15].”
- We have added details regarding new techniques related to ICH in the Discussion. Please see Page 11, Line 528-539
“With the advent of augmented reality/mixed reality techniques, new frameless navigation product, like Brainlab system was recently used as a faster and more feasible technique for minimally invasive hematoma evacuation in ICH patients. These were proven to have a close accuracy for intracranial puncture when compared with the current conventional frame-based stereotactic puncture in an intracerebral hemorrhage phantom model[34]. In our study, we used both Leksell Vantage stereotactic frame system and Brainlab frameless stereotactic system for hematoma drainage. No statistical difference was found in both short- and long- term outcomes, as well as rebleeding, which also indicated the feasibility of Brainlab system in the minimally invasive stereotactic catheter aspiration. However, more studies with large patients cohort are encouraged to validate the accuracy and efficiency of the Brainlab system in the use of minimally invasive stereotactic catheter aspiration after ICH.”
- Future directions of ICH surgery are now mentioned in the conclusion. Please see Page 11, Line 558-562
“which warrants further study of large, prospective, randomized trials for evaluating the best time window to perform the minimally invasive stereotactic catheter ICH evacuation. A better understanding of the surgical time window for stereotactic catheter ICH evacuation and evaluation of preoperative rebleeding risk may benefit patients with ICH.”
- The manuscript is weak in terms of data collection and representation.
A: Thank you for your comment. We have added shortcomings regarding data collection in our Limitation section. Please see Page 11, Line 540-543
“Firstly, it is a retrospective, single-center analysis relying on medical records, and those patients who were lost to follow-up were excluded, which may not accurately reflect the patient conditions and bring some potential bias for the results.”
Reviewer 2 Report
This is a single-center, retrospective cohort study with the aim to describe the prognosis of patients receiving stereotactic catheter aspiration therapy for ICH.
Although this study is of potential interest, several issues exist and should be further assessed by the authors.
1. Language is of suboptimal quality and makes the manuscript hard to follow. Major language editing is needed.
2. Abstract: “Conclusions: Lobar-ICH and 36 rebleeding independently predicted poor outcome in patients with stereotactic catheter ICH evacuation.” - this statement is only partly true. Other factors were also shown to be correlated with mRS 4-6. Why did the authors choose to present only this two in the conclusions of the abstract?
3. Excluding follow-up is a significant bias of this study and should be further commented in the limitation section.
4. The authors may consider describing their center, its workload, its patient volume, the cover district, since having performed stereotactic catheter aspiration therapy in more than 100 patients in only 7 months is an issue of interest!
5. Has a sample size estimation been performed? Or was there another reason that led the authors to terminate the patients’ inclusion in July 2021?
6. Outcomes of interest should be thoroughly presented in the methods section. Otherwise, it seems that the majority of the many analyses that have been performed is strictly exploratory.
7. Did all continuous variables present normal distribution? In the case of skewed distribution, median with interquartile range should be presented rather than mean with sd.
8. “Early hematoma evacuation presented with better outcomes both at 3 months and 1 year after discharge.” – In fact, considering the statistical analysis that was performed, early hematoma evacuation was associated with a lower likelihood of bad outcomes. It is not equal (statistically speaking) to better outcomes though. The authors should consider revising this statement and similar ones throughout the whole text. Importantly, good outcomes was not even part of the outcomes of interest, as presented in the methods section.
Author Response
Reviewer 2
- Language is of suboptimal quality and makes the manuscript hard to follow. Major language editing is needed.
A: Thank you for your comment. We have consulted an editing service to improve our manuscript’s language.
- Abstract: “Conclusions: Lobar-ICH and rebleeding independently predicted poor outcome in patients with stereotactic catheter ICH evacuation.” - this statement is only partly true. Other factors were also shown to be correlated with mRS 4-6. Why did the authors choose to present only this two in the conclusions of the abstract?
A: Thank you for your question. We agree this statement is only partly true. We have changed this sentence to better reflect our attention. Please see Page 1, Line 36-38
“Lobar-ICH and rebleeding independently predicted both poor short- and long-term outcomes in patients with stereotactic catheter ICH evacuation”
- Excluding follow-up is a significant bias of this study and should be further commented in the limitation section.
A: Thank you for your comment. This limitation was added. Please see Page 11, Line 540-543
“Firstly, it is a retrospective, single-center analysis relying on medical records, and those patients who were lost to follow-up were excluded, which may not accurately reflect the patient conditions and bring some potential bias for the results.”
- The authors may consider describing their center, its workload, its patient volume, the cover district, since having performed stereotactic catheter aspiration therapy in more than 100 patients in only 7 months is an issue of interest!
A: Thank you for your comment. We have added description of our center in the Methods section. Please see Page 2, Line 95-97
“The neurosurgery center of our hospital is the largest center in Zhejiang Province, which performs more than 9000 surgeries every year, with more than 300 surgical ICH evac-uations every year.”
- Has a sample size estimation been performed? Or was there another reason that led the authors to terminate the patients’ inclusion in July 2021?
A: Thank you for your question. We have two surgical teams for stereotactic catheter ICH aspiration. All data reviewed for patients with spontaneous ICH treated with stereotactic catheter aspiration between January 1, 2021, and July 31, 2021, were conducted by one surgical team led by Dr. Fengqiang Liu. Another surgical team was responsible for stereotactic catheter ICH aspiration from August 1, 2021, to December 31, 2021. To avoid potential bias carried by different surgical groups, only patients treated by the team of Dr. Fengqiang Liu between January 1, 2021, and July 31, 2021, were analyzed.
- Outcomes of interest should be thoroughly presented in the methods section. Otherwise, it seems that the majority of the many analyses that have been performed is strictly exploratory.
A: Thank you for your suggestion. The outcome assessment was detailed in the section of Outcome Assessment. Please see Page 3, Line 170-175
“Functional outcomes assessed by the modified Rankin Scale (mRS)[22] were followed by telephone, hospital visits, telephone calls or clinic visits at 3 months (short-term) and 1 year (long-term) after discharge. mRS scores ranging from 0-3 were considered good outcomes, ranging from 4-6 were considered poor outcomes[21]. In addition, 30-day mortality, NICU length of stay (LOS), and hospital LOS were also reviewed. Rebleeding was defined as ICH growth >30% or >6mL from the baseline ICH volume[3].”
- Did all continuous variables present normal distribution? In the case of skewed distribution, median with interquartile range should be presented rather than mean with sd.
A: Thank you for your suggestion. Those continuous variables with skewed distribution were changed to median with interquartile range, please see Table 4. All other continuous variables are presented as normal distributions.
- “Early hematoma evacuation presented with better outcomes both at 3 months and 1 year after discharge.” – In fact, considering the statistical analysis that was performed, early hematoma evacuation was associated with a lower likelihood of bad outcomes. It is not equal (statistically speaking) to better outcomes though. The authors should consider revising this statement and similar ones throughout the whole text. Importantly, good outcomes was not even part of the outcomes of interest, as presented in the methods section.
A: Thank you for your advice. We agree with your opinion, these statements have been changed to “early hematoma evacuation presented a lower likelihood of poor outcomes both at 3 months and 1 year after discharge” in the MS.
Round 2
Reviewer 1 Report
In the revised version, author has resolved the issues we had. I dont have any other comment for this manuscript.
Reviewer 2 Report
The issues raised in the first round of review have been addressed.